# Structural Aspects of LIMK Regulation and Pharmacology

**DOI:** 10.3390/cells11010142

**Published:** 2022-01-02

**Authors:** Deep Chatterjee, Franziska Preuss, Verena Dederer, Stefan Knapp, Sebastian Mathea

**Affiliations:** 1Structural Genomics Consortium, Buchmann Institute for Molecular Life Sciences, Max-von-Laue-Str 15, 60438 Frankfurt am Main, Germany; chatterjee@nmr.uni-frankfurt.de (D.C.); Preuss@pharmchem.uni-frankfurt.de (F.P.); Dederer@pharmchem.uni-frankfurt.de (V.D.); knapp@pharmchem.uni-frankfurt.de (S.K.); 2Institute for Pharmaceutical Chemistry, Johann Wolfgang Goethe-University, Max-von-Laue-Str 9, 60438 Frankfurt am Main, Germany

**Keywords:** LIMK1, LIMK2, kinase, phosphorylation, small-molecule inhibitor, catalytic mechanism, cofilin, actin cytoskeleton dynamics

## Abstract

Malfunction of the actin cytoskeleton is linked to numerous human diseases including neurological disorders and cancer. LIMK1 (LIM domain kinase 1) and its paralogue LIMK2 are two closely related kinases that control actin cytoskeleton dynamics. Consequently, they are potential therapeutic targets for the treatment of such diseases. In the present review, we describe the LIMK conformational space and its dependence on ligand binding. Furthermore, we explain the unique catalytic mechanism of the kinase, shedding light on substrate recognition and how LIMK activity is regulated. The structural features are evaluated for implications on the drug discovery process. Finally, potential future directions for targeting LIMKs pharmacologically, also beyond just inhibiting the kinase domain, are discussed.

## 1. LIMKs Regulate Actin Dynamics

The cellular actin cytoskeleton is a permanent construction site. Its filaments constantly grow or shrink, form branches, or get disrupted, attach to membranes or to cargo vesicles. Despite being highly dynamic, the filaments have to withstand mechanical forces [1]. The balance between dynamics and stability is adjusted by accessory proteins such as actin-depolymerizing factors (ADFs). Binding of ADFs destabilizes the filaments and leads to severing and disassembly of the affected filament sections [2]. The ADF activity, however, is regulated by phosphorylation (Figure 1A). When phosphorylated by LIM domain kinases (LIMKs), the ADFs are inactive [3]. Dephosphorylation by Slingshot homolog 1 (SSH1) restores ADF activity [4]. Like this, the LIMKs and SSH1 control the dynamics of the actin cytoskeleton, with significant roles in physiology and disease [5]. There are two LIMKs expressed in humans, namely LIMK1 and LIMK2, both containing a kinase domain in their C terminus (Figure 1B). To add another layer of complexity, several LIMK splicing variants were identified on the mRNA and protein level [6]. This review explores structural aspects of LIMK catalytic activity and regulation and discusses how a better understanding of LIMK features enables pharmacological targeting of the actin cytoskeleton plasticity.

## 2. The Conformational Space of the LIMK Kinase Domain

### 2.1. Basic Features of the Kinase Fold

Similar to other protein kinases [7], the LIMK kinase domains are composed of two lobes. The N lobe with its curved antiparallel β sheet is smaller than the globular C lobe with its α helices. The ATP-binding cleft between the two lobes is surrounded by conserved sequence motifs that are indispensable for the kinase catalytic activity: The G-rich loop forms a lid to a potentially bound ATP molecule and shields it from the solvent (consensus sequence GXGXXG, LIMK1 sequence GKGCFG). The VAIK motif (MVMK in LIMK1) contributes to forming the hydrophobic cage for the ATP adenine and contains the lysine that links the β3 strand to the αC helix via the K368 E384 salt bridge. The catalytic loop orients the potentially bound ATP molecule for catalysis and harbours the catalytic aspartate that interacts with the potential phosphoacceptor residue (consensus sequence HRDXKXXN, LIMK1 sequence HRDLNSHN). And finally, the DFG motif in the base of the activation loop can switch between the DFG-in and the DFG-out conformation. Only in the DFG-in conformation, the kinase is capable of binding to ATP and of catalysing the phosphoryl transfer reaction. The LIMK kinase domain is structurally well explored [8,9,10]. To date, nine structure models have been deposited to the PDB (Table 1). Diverse small molecules are bound to the LIMK active site, ranging from nucleotides to type-1 and type-3 kinase inhibitors. Accordingly, the structure models differ substantially in conformation.

### 2.2. The Active LIMK Kinase Conformation

When having bound a nucleotide such as ADP or ATP-γ-S, the LIMKs adopt the canonical active kinase conformation (Figure 1C). The ATP adenine is surrounded by a hydrophobic cage, while the phosphates are held in position by polar LIMK residues such as asparagine N465 and aspartate D478 (Figure 2C). The conserved salt bridge between the VAIK lysine and the αC glutamate is formed, resulting in a firmly attached αC helix. Furthermore, the G-rich loop serves as a lid enclosing the bound nucleotide and shielding it from the solvent. The base of the activation loop adopts the DFG-in conformation with the phenylalanine sidechain deeply buried in a hydrophobic pocket in the back of the ATP-binding site and leaving the aspartate side chain exposed to interact with the ATP phosphates. Interestingly, the LIMKs have an asparagine residue in the HRD + 2 position instead of the canonical lysine. This potentially has an impact on the apparent K_m_ for ATP. Both the catalytic and regulatory spines [11] are formed, further stabilizing the active kinase conformation (Figure 1C). Taken together, all residues are perfectly positioned for catalysis even in the absence of activation loop phosphorylation. Notably, being looked at from the side (Figure 2A), the LIMK kinase domains appear unusual shallow, indicating the unique substrate recognition and catalysis mechanisms (discussed in a later section).

### 2.3. Inactive LIMK Kinase Conformations

The binding of small-molecule inhibitors often distorts the kinase fold, stabilizing inactive kinase conformations instead. All LIMK inhibitors reported to date bind to the ATP pocket [8,10] and reshape neighbouring structure elements, mainly in the kinase N lobe. This is particularly evident for the G-rich loop, which can appear rotated or even detached upon inhibitor binding (Figure 1D). Several inhibitors displace the αC helix and push it outwards [8,10]. As a consequence, the aforementioned K-E salt bridge is broken (Figure 1E). Furthermore, the swung out αC helix gives way to the DFG flip. In the active conformation, the DFG phenylalanine resides in a hydrophobic pocket formed by the αC and αE helices and the catalytic loop. The DFG flip removes the phenylalanine from the back pocket and installs it in the active site (Figure 1F). Some inhibitors occupy the empty back pocket, thus stabilizing the LIMKs in the DFG-out conformation (type-1, type-2 and type-3 inhibitors will be discussed in a later section). Notably, even in the absence of inhibitors, the LIMK kinase domains can adopt both, the DFG-in and the DFG-out conformation, suggesting a low energy barrier between both conformations [10]. We believe that the existing LIMK kinase structure models cover most of its conformational space. Similar to other kinases, the molecular dynamics include the rotation of the G-rich loop, the swinging out of the αC helix, the isomerization of the DFG motif to the out conformation, the detachment of the activation loop, and a general distortion of the N-lobal β sheet relative to the more rigid C lobe.

### 2.4. Kinase Activation by T508 Phosphorylation

A wide range of extracellular and intracellular events modulate actin cytoskeleton plasticity. The respective signalling network is complicated, but well established [12,13]. Several signalling cascades converge at the LIMKs, making them a major integration node. Four kinases have been described to directly regulate the LIMK activity by phosphorylation, namely PAK1/4 [14], ROCK1 [15] and MRCKα [16]. While the physiological significance differs between the signalling cascades, the mechanism triggered by the phosphorylation event is identical.

The phosphorylated residue is threonine T508 within the LIMK1 activation loop (or T505 in LIMK2, respectively) [17]. Upon phosphorylation, pT508 forms a stable salt bridge with arginine R483, which corresponds to the DFG + 3 position (Figure 1G). The impact of the pT-R salt bridge formation is twofold: On the one hand, the DFG motif is sterically hindered from adopting the DFG-out conformation, and the entire N lobe is stabilized in the active conformation. On the other hand, the activation loop is oriented to attach to the C lobe, thus contributing to a docking interface for protein substrates (shown in yellow in Figure 2D). As a consequence, the LIMK kinase activity is increased dramatically [10]. An interesting aspect of LIMK activation is observed with the upstream kinase PAK4, which prefers phosphorylating its substrates at serine residues [18]. Accordingly, all validated PAK4 substrates except for LIMK1 are phosphorylated at serine residues, and consequently the LIMK1 variant T508S is a much better substrate for PAK4 than LIMK1 WT. This represents an example of the physiological phosphorylation of a disfavoured kinase substrate [18].

Importantly, LIMK activation loop phosphorylation also has an impact on inhibitor design. The phosphorylated form of LIMK1 adopts the DFG-in conformation and is catalytically active. Type-2 and type-3 kinase inhibitors, however, bind exclusively to the DFG-out conformation. Accordingly, these compounds tightly bind to the non-phosphorylated (inactive) LIMKs, but their interaction with the phosphorylated (active) LIMKs is much weaker. Therefore, the suitability of type-2 and type-3 kinase binders to inhibit the cellular LIMK activity must be critically evaluated.

## 3. The Unusual LIMK Catalytic Mechanism

### 3.1. Catalytic Mechanism and Fidelity Control in Conventional Kinases

To appreciate how special the LIMK catalytic mechanism is, we need to look into conventional kinases first [19]. They usually phosphorylate their substrate proteins in flexible and easily accessible loops. The phosphoacceptor loop binds to the groove between the kinase N and C lobes. In the course of this binding event, the phosphoacceptor residue is positioned next to the ATP γ phosphate, and catalysis can occur. As an example, the ternary complex formed by the kinase AKT2, a GSK3-derived substrate peptide and the co-substrate analogue AMP-PNP is depicted in Figure 2B [20].

According to the outlined mechanism, there are three levels of substrate fidelity control [19]. The first level involves protein interaction modules on the kinase surface. They are located distal to the active site, either in the kinase domain, in flanking domains or even in associated scaffolding proteins, and form docking interfaces for the substrate protein. Binding is mediated by parts of the substrate protein that are again distal to the phosphoacceptor. The interaction of kinase and substrate protein orients the substrate and brings the phosphoacceptor loop close to the phosphoacceptor loop binding groove.

The phosphoacceptor loop includes the phosphoacceptor and flanking residues (about 10 residues altogether). Its linear sequence constitutes the second level of fidelity control because it has to match the phosphoacceptor loop binding groove to allow for the formation of the transient kinase-substrate complex. In our example AKT2, the arginine residues in the –5 and –3 positions (counting from the phosphoacceptor residue) are particularly important—they form salt bridges with the binding groove. (Figure 2B). In addition, there are main chain interactions between the phosphoacceptor loop and the kinase activation loop. However, most kinases recognize very diverse linear motifs, and conversely, the identification of the particular kinase that physiologically phosphorylates a given linear motif is challenging [21]. Nevertheless, information on preferred linear motifs often allows excluding potential phosphoacceptors.

The third level of fidelity control bases upon the length of the phosphoacceptor side chain. Kinases with a deeper pocket preferably phosphorylate the long tyrosine side chains, while kinases with a shallower pocket rather phosphorylate short side chains such as serine and threonine. This is an intrinsic feature of the conventional kinase catalytic mechanism with the rigid positioning of ATP and the phosphoacceptor loop (Figure 2B) [22]. In the LIMKs, both the catalytic mechanism and the fidelity control differ substantially from conventional kinases.

### 3.2. LIMK Substrate Recognition and ‘Rock-and-Poke’ Mechanism

The LIMK1 kinase domain appears shallow when looked at from the side (Figure 2A). Its αG helix is shifted towards the C terminus by about 13 residues in comparison to other kinases, giving space for the αF αG loop and the activation loop to form a docking interface for substrates (Figure 2D). ADFs such as CFL1 bind this docking interface with their anchor helix [9]. The CFL1 residue K112 plays a crucial role in binding as demonstrated by comparing CFL1 WT and CFL1 K112A as LIMK1 substrates [10]. The interaction corresponds to what was described above as the first level of fidelity control in conventional kinases since the LIMK1 docking interface is distal to the kinase active site, and the CFL1 anchor helix is distal to the phosphoacceptor. However, unexpectedly, this is the only interaction between the kinase and the substrate. CFL1 does not contain any phosphoacceptor loop—the residues flanking the phosphoacceptor do not interact at all with the kinase. The phosphoacceptor serine S3 is positioned in the very N terminus of the CFL1 protein (methionine M1 is posttranslationally removed). As a consequence, there is no second level of fidelity control in LIMKs. 

In the available structure models of the ternary complex composed of the kinase LIMK1, the substrate protein CFL1 and the co-substrate analogue ATP-γ-S, the CFL1 orientations in the complex differ. More precisely, there is only one orientation for the anchor helix, but the rest of CFL1 is rotated by 12° (indicated by an arrow in Figure 2D) [9,10]. Obviously, the ends of the anchor helix serve as hinges, allowing for a rocking movement in the CFL1 protein, with the phosphoacceptor poking into the kinase active site until a constructive orientation is achieved and catalysis occurs. This mechanism was referred to as the ‘rock-and-poke’ mechanism and is unique for the LIMKs [10].

A surprising implication of the unusual mechanism is that LIMK1 is capable of phosphorylating both, serine and tyrosine residues, which makes it a dual-specificity kinase, while its activity towards threonine residues is low [10,23]. Obviously, specificity is here not defined by the length of the side chain (as discussed above for conventional kinases), but by steric shielding of the hydroxy group. Accordingly, the third level of fidelity control in LIMKs is based on the accessibility of a hydroxy group in the very N terminus of the substrate protein. A similar discrimination between serine and threonine phosphoacceptors was observed for the kinase PAK1 [24]. Voluminous residues such as phenylalanine in the DFG + 1 position sterically shield threonine from being phosphorylated. An equivalent role of the DFG + 1 leucine in LIMK1 can be hypothesised but has not been shown experimentally to date. The human kinome comprises several dual-specificity kinases. The physiological phosphorylation of threonine and tyrosine residues is a well-established feature of kinases such as MEK6 [25]. The LIMKs phosphorylate their substrates mainly at serine residues, but interestingly, also the cellular phosphorylation of an internal tyrosine residue has been described [26]. 

The mode of CFL1 binding to actin filaments is similar to its binding to the LIMK kinase domain [27]. Both interactions are driven by the anchor helix. A possible mechanism of action for the cellular LIMKs is that their N-terminal LIM domains glue them to specific actin filaments. Potentially present CFL1 then binds the LIMKs instead of the actin filaments, is phosphorylated and thereby inactivated. In this manner, the LIMKs might protect specific actin filaments from disassembly.

### 3.3. Role of the LIM Domains and the PDZ Domain

In addition to the well-characterized kinase domain, LIMKs contain several protein interaction modules in their N termini (Figure 1B). LIM domains (the name is an acronym from the proteins **L**IN-11, **I**sl-1 and **M**EC-3) are stabilized by zinc ions and frequently occur as tandems or in even higher numbers [28]. Several LIM domains interact with the actin cytoskeleton. Examples of this canonical interaction are the LIM domain-containing proteins paxillin and zyxin [29]. Even though direct experimental evidence is lacking, it has been hypothesized that the LIM domains anchor the LIMKs to the actin cytoskeleton. Other potential binders of the LIMK LIM domains include PARD3 [30] and NR4A2 [31]. Notably, the structure of the second LIM domain of LIMK2 has been elucidated by NMR (PDB ID 1X6A).

PDZ domains (the name is derived from the proteins **P**SD95, **D**lg1 and **z**o-1) are globular protein interaction modules that bind to short linear motifs in the C terminus of their protein ligands [32,33]. PDZ domains are known to bring together dynamic signalling complexes as described for PICK1 [34] and PSD-95 [35]. The LIMK PDZ domains might be involved in similar processes. Specifically, NF1 [36] and nischarin [37] were described to be cellular interaction partners of the LIMK PDZ domains. The structure of the PDZ domain from mouse LIMK2 has been solved by NMR (PDB ID 2YUB). In addition to a potential role in defining the subcellular localization and in complex formation, the N-terminal protein interaction modules may also regulate the activity of the C-terminal kinase domain. In other kinases, such N-terminal domains act as intramolecular inhibitors as described for PAK1 [38], or they recruit specific protein substrates as observed for ULK3 [39,40]. At first glance, the long unstructured linker between the PDZ and kinase domains in the LIMKs (about 70 residues) rather indicates that the N and C termini do not interact physically. However, there are reports demonstrating this interaction [41]. Furthermore, the N-terminal LIM domains were suggested to inhibit the kinase domain [42,43]. An additional mechanism to control LIMK activity is to phosphorylate the PDZ-kinase linker as shown in cell-culture experiments for MAPKAPK2 [44] and Aurora A [45]. Early work has also pointed to HSP90 as a regulator of LIMK dimerization, stability and activity [46]. This list is by no means complete. A plethora of LIMK substrates, interactors and activators have been published during the past 25 years. To identify the most significant physiological LIMK pathways, also resolved in terms of cell type and developmental stage, the inhibitors introduced in the next paragraphs might be useful.

## 4. Pharmacological Targeting of LIMKs

### 4.1. LIMKs Are Involved in Human Disease

In line with its fundamental role in regulating actin dynamics [3], changes in LIMK signalling were frequently linked to human pathology. Recent examples include amyotrophic lateral sclerosis [47], fragile-X mental retardation syndrome [48], neurofibromatosis type 2 [49], colorectal cancer progression [50] and castration-resistant prostate cancer [51] (a critical target evaluation is not in the scope of this review). The suggested disease rationales differ in terms of cause and signalling pathway. Interestingly, however, they all involve hyperactive LIMKs, indicating that patients will potentially benefit from the inhibition of LIMK kinase activity. From the pharmacology perspective, this is good news since it is technically more feasible to develop kinase inhibitors than kinase activators. Indeed, several inhibitors have been developed to date. In the last sections of the review, general aspects of LIMK inhibitor development are discussed.

### 4.2. LIMK Inhibitors Binding to the Active Site

It is common practice to target kinases with small molecules that bind to the active site. To date, 71 of such molecules have been approved as drugs by the FDA, mainly for the treatment of cancer and inflammatory disorders [52]. According to their exact position in the cleft between the kinase N and C lobes, the ligands can be classified as type-1, type-2 or type-3 inhibitors (Table 2). Notably, extended naming schemes with more detailed differentiation of inhibitor binding modes have been suggested [53]. Interestingly, the LIMKs have been targeted by inhibitors of all three main binding types, reflecting the LIMK conformational plasticity discussed above [8,10,54]. In addition to the front pocket next to the hinge region and the hydrophobic back pocket, several elements from the LIMK active site can contribute to inhibitor binding, most prominently the G-rich loop backbone, the VAIK motif lysine, the gatekeeper threonine, the HRD motif histidine and the DFG motif aspartate. This is by no means unusual for a kinase. Nevertheless, good selectivity within the kinome was achieved for the LIMK inhibitors LIMKi3 [54] and Ligand 22 [8]. The binding modes of several inhibitors are compared in Figure 3.

Phosphorylation of the threonine residue T508 in the activation loop locks LIMK1 in the active DFG-in state [10]. This state is preferably bound by type-1 inhibitors, while type-2 and type-3 inhibitors exclusively bind to the inactive DFG-out state. As a consequence, the affinities of type-2 and type-3 inhibitors to phosphorylated LIMK1 are dramatically reduced in comparison to the non-phosphorylated protein. On top of this, type-3 inhibitor binding to non-phosphorylated LIMK1 has no effect on LIMK1 phosphorylation by upstream kinases such as PAK1 (unpublished observations). Depending on the biological setting, type-2 and type-3 inhibitors can be employed to target the non-phosphorylated LIMKs. In any case, the phosphorylation-state specificity needs to be considered when evaluating LIMK inhibitors in a cellular environment.

### 4.3. Inhibitors Interfering with Substrate Recognition

The rational targeting of kinase domains via binding pockets different from the active site is challenging and requires a profound understanding of the particular catalytic mechanism [52]. Examples of such inhibitors that bind the kinase domain and remodel its protein interaction network are the allosteric MEK1 inhibitors trametinib and cobimetinib [55]. In principle, the LIMKs are ideal targets for the development of allosteric inhibitors since their catalytic mechanism is understood in detail. The CFL1 anchor helix can serve as a template for the design of inhibitory peptides. As a proof-of-concept experiment, these peptides can be probed in a LIMK1 activity assay using CFL1 as a substrate. However, with potent and selective competitive inhibitors already in place, it needs to be carefully evaluated whether this is a worthwhile endeavour.

### 4.4. PROTACs to Induce LIMK Degradation

While active-site inhibitors solely decrease the catalytic activity of the target kinase, PROTACs lead to ubiquitination and degradation of the protein in a cellular environment. When applied to LIMKs, both strategies are expected to have different physiological impacts since the LIMKs with their modular architecture may also act as scaffolds. An assessment of several promiscuous PROTACs demonstrated that the PROTAC strategy is suitable for the LIMKs—they were even classified as ‘highly degradable targets’ [56]. Another less kinase-centred approach is to target the LIMK PDZ domain with PROTACs. However, developing selective PDZ warheads is expected to be challenging [57].

### 4.5. Outlook—Isoform-Specific LIMK Inhibitors

The LIMK1 and LIMK2 kinase domains share high sequence similarity (71% identical). In the active site and the substrate docking interface there are barely any differences. More variety is found in the back of the C lobe (β7 β8 loop, αE and αI helices), a functionally unexplored part of the LIMKs. Despite the conserved active site, LIMK1 and LIMK2 differ slightly in substrate specificity [58], and several ATP-competitive inhibitors were identified that bind to LIMK1, but not to LIMK2 [10], indicating that it is indeed possible to develop isoform-specific LIMK inhibitors. An alternative approach to achieve selectivity is to develop covalent inhibitors. LIMK1 contains a cysteine in the G-rich loop (sequence GKGCFG), while LIMK2 has not (sequence GKGFFG). It can be regarded as a proof-of-concept finding that a cysteine in exactly the same position was targeted successfully in the FGFRs and in SRC [59].

Due to their involvement in the same cellular pathway and their overlapping substrate specificity, differences in the physiological roles of both LIMKs are not entirely resolved. Clues come from their tissue expression profiles. While LIMK1 is mainly expressed in the human brain, LIMK2 is widely expressed in all tissues (taken from The Human Protein Atlas). Accordingly, potential adverse effects from LIMK1 inhibition include abnormal synaptic function [60,61,62], impaired platelet activation [63] and reduced osteoblast number [64]. LIMK2 inhibition, however, was reported to impair spermatogenesis [65] and platelet function [66]. This highlights the need for isoform-specific LIMK inhibitors to validate the individual LIMKs as therapeutic targets and, in the next phase, to develop therapeutics with the lowest possible risk of severe adverse effects.

## Figures and Tables

**Figure 1 cells-11-00142-f001:**
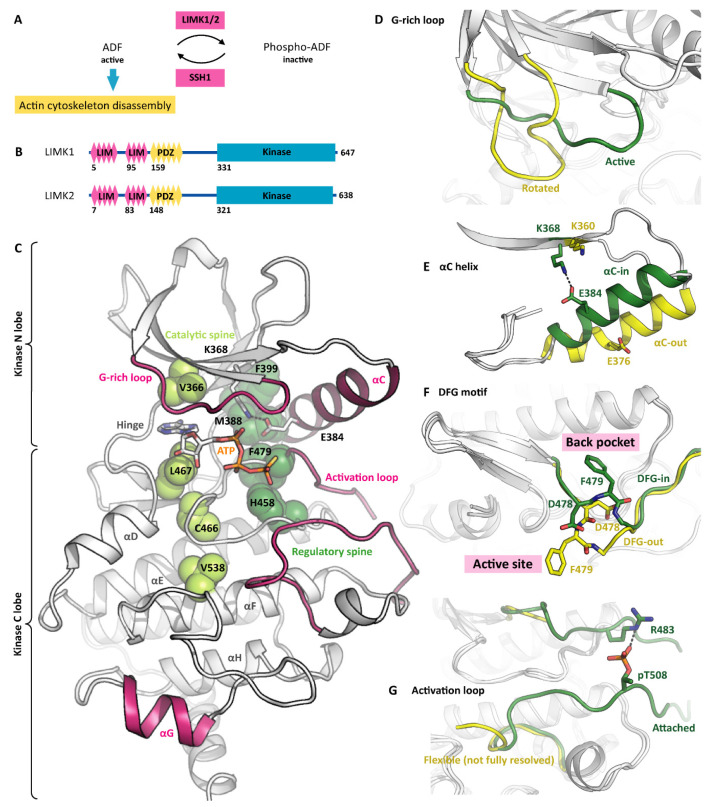
The LIMK conformational space. (**A**) LIMK1/2 and the phosphatase SSH1 control the dynamics of the actin cytoskeleton. ADF—actin depolymerizing factor. (**B**) Both LIMK1 and LIMK2 contain N-terminal protein interaction modules and a C-terminal kinase domain. (**C**) With ATP-γ-S bound, LIMK1 adopts the canonical active-kinase conformation. The catalytic and regulatory spines are fully formed (indicated in green). Please note the G-rich loop enclosing the co-substrate, the αC-in conformation, the attached activation loop, and the unusual orientation of the αG helix (all indicated in pink). PDB ID 5L6W. (**D**) Inhibitor binding can induce a rotated G-rich loop conformation. PDB IDs 5L6W and 5NXD. (**E**) The αC helix is capable of adopting both, the αC-in and the αC-out conformation. PDB IDs 5L6W and 5NXD. (**F**) The DFG motif in the base of the activation loop switches between the DFG-in and the DFG-out conformation. PDP IDs 5L6W and 5NXD. (**G**) Conformational plasticity is also observed in the activation loop, which can either be attached to the C lobe or flexible. Please note that the flexible loop is not fully resolved in the crystal structure. PDB IDs 5HVJ and 5NXC.

**Figure 2 cells-11-00142-f002:**
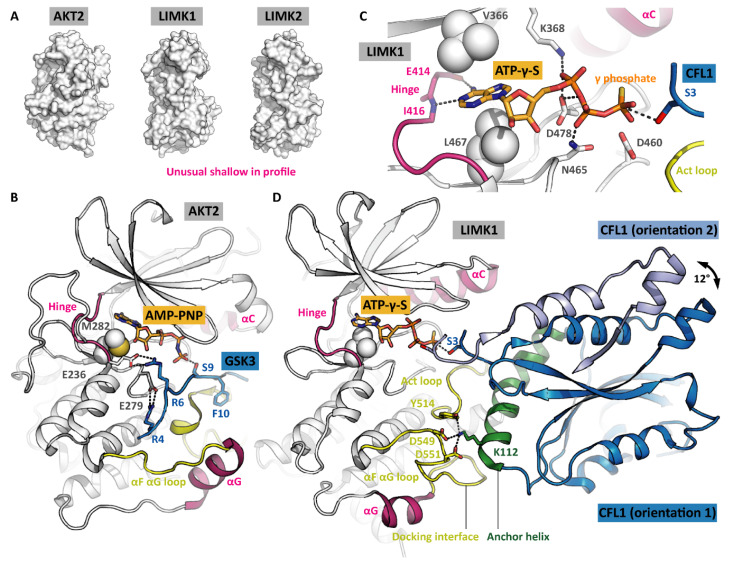
Comparison of kinase-substrate recognition modes. (**A**) Profile views of the AKT2 and LIMK kinase domains to showcase their overall appearance. (**B**) Snapshot of AKT2 with substrate and co-substrate bound in the moment of catalysis. The phosphoacceptor loop is tightly attached to the kinase, positioning the phosphoacceptor residue (serine S9) next to the ATP γ phosphate. PDB ID 1O6K [20]. (**C**) ATP interacts with the LIMK1 hinge, its adenine rings are sandwiched by hydrophobic residues, and its phosphates are oriented by polar residues. (**D**) LIMK1 with substrate and co-substrate bound, again in the moment of catalysis. The substrate protein attaches to the kinase solely with its anchor helix—the residues flanking the phosphoacceptor residue (serine S3) do not interact with the kinase. The arrow indicates the rocking movement that allows the phosphoacceptor to poke into the active site. The two CFL1 orientations are taken from independent crystal structures. Parts of CFL1 orientation 2 are omitted for clarity. PDB IDs 5HVJ and 5L6W [9,10].

**Figure 3 cells-11-00142-f003:**
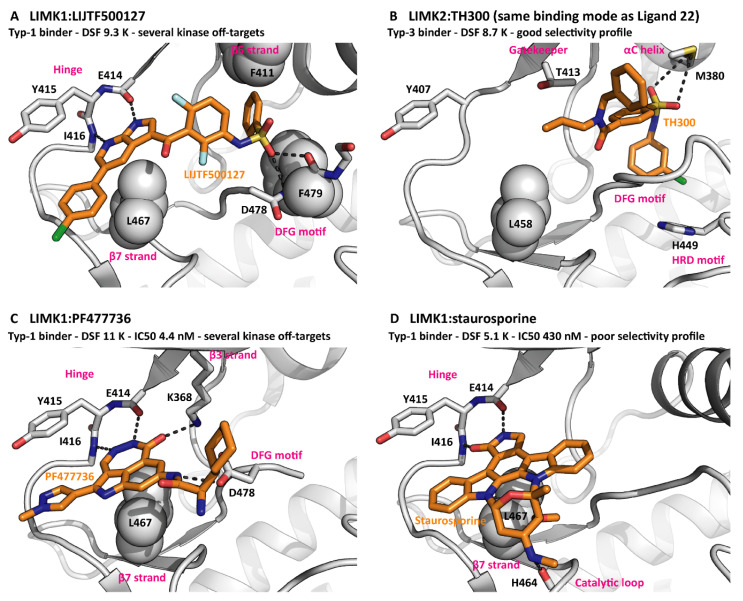
Inhibitors bound to LIMKs. (**A**) The type-1 binder LIJTF500127 spans from the hinge to a pocket between the β5 strand and the DFG motif. PDB ID 7ATS. (**B**) The type-3 binder TH300 mainly exhibits hydrophobic interactions with kinase elements surrounding the back pocket. PDB ID 5NXD. (**C**) PF477736 occupies the front pocket only but interacts with charged residues from the β3 strand and the DFG motif. PDB ID 5NXC. (**D**) The pan-kinase inhibitor staurosporine binds due to its disc shape and its hinge interactions. PDB ID 3S95. The G-rich loops are omitted for clarity reasons.

**Table 1 cells-11-00142-t001:** LIMK kinase domain structure models.

Protein	Boundaries	Ligand	Ligand Type	Space Group	PDB ID
LIMK1	330–637	Staurosporine	Type-1	C 2 2 21	3S95 [10]
LIMK1	330–637	PF477736	Type-1	C 2 2 21	5NXC [10]
LIMK1	330–637	LIJTF500127	Type-1	P 61 2 2	7ATS
LIMK1	330–637	LIJTF500025	Type-3	P 21	7ATU
LIMK2	330–632	Ligand 22	Type-3	P 21	4TPT [8]
LIMK2	330–632	TH300	Type-3	P 21	5NXD
LIMK1 D460N	329–638	AMP-PNP	Nucleotide	P 21	5HVJ [9]
LIMK1 D460N	329–638	ADP	Nucleotide	P 21 21 21	5HVK [9]
LIMK1	330–637	ATP-γ-S	Nucleotide	P 32 2 1	5L6W [10]

**Table 2 cells-11-00142-t002:** Classification of LIMK active-site inhibitors.

Ligand Type	Hinge Interaction	Back Pocket Occupation	Example Inhibitor	Kinome Selectivity
Type-1	+	−	LIMKi3 [54]	High
Type-2	+	+	Rebastinib [10]	Low
Type-3	−	+	Ligand 22 [8]	High

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
