# Peer review of "Structural Aspects of LIMK Regulation and Pharmacology"

_cells, 2022, doi:10.3390/cells11010142_

Round 1

Reviewer 1 Report

In this paper entitled “Structural Aspects of LIMK Regulation and Pharmacology”, Chatterjee et al. make a detailed overview of the atypical structural properties of the LIM kinases. It is the first time that such data are collected together, as most of the structures have been deposited in the Protein Database without any associated publications. This paper is a great contribution to the field.

However, some points have to be addressed to make it proper for publication.

  1. Paragraph 1.1. “The active kinase conformation” is well detailed. Nevertheless, for structure non-specialists it is hard to follow. It would be very useful for a better and easier understanding of this part to start with a general description of kinase active site and conformation: what are VAIK lysine, G-rich loop, aC helix, DFGin/out, catalytic and regulatory spines?… from a structural and functional point of view.
  2. In figure 1, a zoom on the ATP binding area would be very useful to highlight and annotate in more detail the residues involved into this interaction: hydrophobic cage, Asn478, Asp478, DFG lateral chain positions.
  3. Authors claim that the “LIMK kinase domains appear unusual shallow”. From Figure 1C, it is difficult to realise that whereas it is a very important feature for LIMKs. Would it be possible to make an image where LIMK kinase domain and any other kinase domain are superimposed to clearly see the difference?
  4. Would it be possible to have an image with the DFG flipping upon inhibitor binding? (paragraph 1.2)
  5. Kinases phosphorylating LIMKs on Thr508/505 are ROCK, PAK and MRCKa (Sumi et al., 2001, JBC, 276, 25, 23092-23096). BMPRII phosphorylates LIMKs but it is not known on which amino acid.
  6. Paragraph 1.3., Figure 2B does not illustrate very well what is depicted in the text (salt bridge between P-T508/R483, DFG out conformation, …would it be possible to have a dedicated figure illustrating in detail these points?
  7. Paragraph 2.1., in Figure 2A R4 and R6 are labelled whereas in the text R3 and R5 are mentioned. Are there any mistakes?
  8. Paragraph 2.2, what is aF loop?, what is its role for the kinase structure and activity?
  9. Paragraph 2.2. Is it possible to discuss a bite more about LIMK dual specificity (Ser/Tyr)? Is it common for other kinases? How may this feature be explained?
  10. Paragraph 2.3., please mention the structures of LIMK LIM and PDZ structures obtained by NMR and deposited in the PDB. Maybe comments about these structures are necessary?
  11. In paragraph 2.3., several claims are not exact. Edwards and Gill (1999, JBC 274, 16, 11352) showed that LIMK LIM domains negatively regulate its kinase activity, and Hiraoka et al (1996, FEBS Letters, 399, 117) showed an interaction between N and C-terminal domains of LIMKs. Several partners of LIMKs involving their LIM and PDZ domains have been identified: Par3 interacts with the LIM domain of LIMK2 (Chen and Macara, 2006, JCB 172,671), Nurr1 interacts with the LIM domain of LIMK1 (Sacchetti et al., NAR 2006, 34, 19, 5515-5527), the SecPH domain of Nf1 interacts with the PDZ of LIMK2 (Vallee et al, 2012 PLOS One, 7, 10 e47283), Nischarin interacts with the PDZ domain of LIMK1 (Ding et al., MCB 2008, 28, 11, 3742), …

Author Response

Dear reviewer --

Thank you very much for your comments / suggestions! This has allowed us to correct several mistakes and to add some important aspects of LIMK biology. Please find below a detailed reply to your review.

Kind regards

Chatterjee et al.

Reviewer 1

In this paper entitled “Structural Aspects of LIMK Regulation and Pharmacology”, Chatterjee et al. make a detailed overview of the atypical structural properties of the LIM kinases. It is the first time that such data are collected together, as most of the structures have been deposited in the Protein Database without any associated publications. This paper is a great contribution to the field.

However, some points have to be addressed to make it proper for publication.

  1. Paragraph 1.1. “The active kinase conformation” is well detailed. Nevertheless, for structure non-specialists it is hard to follow. It would be very useful for a better and easier understanding of this part to start with a general description of kinase active site and conformation: what are VAIK lysine, G-rich loop, aC helix, DFGin/out, catalytic and regulatory spines?… from a structural and functional point of view.

We have added a paragraph to describe general kinase properties.

  1. In figure 1, a zoom on the ATP binding area would be very useful to highlight and annotate in more detail the residues involved into this interaction: hydrophobic cage, Asn478, Asp478, DFG lateral chain positions.

Another panel was added as Figure 2C to showcase the ATP-binding mode.

  1. Authors claim that the “LIMK kinase domains appear unusual shallow”. From Figure 1C, it is difficult to realise that whereas it is a very important feature for LIMKs. Would it be possible to make an image where LIMK kinase domain and any other kinase domain are superimposed to clearly see the difference?

Another panel was added as Figure 2A to illustrate the overall appearance of the LIMKs.

  1. Would it be possible to have an image with the DFG flipping upon inhibitor binding? (paragraph 1.2)

Yes – it is Figure 1F now.

  1. Kinases phosphorylating LIMKs on Thr508/505 are ROCK, PAK and MRCKa (Sumi et al., 2001, JBC, 276, 25, 23092-23096). BMPRII phosphorylates LIMKs but it is not known on which amino acid.

Thank you – this is corrected now.

  1. Paragraph 1.3., Figure 2B does not illustrate very well what is depicted in the text (salt bridge between P-T508/R483, DFG out conformation, …would it be possible to have a dedicated figure illustrating in detail these points?

Figure 1C was corrected to show the DFG phenylalanine – Figure 1G  was modified to better illustrate the salt bridge – Figure 1F was added to illustrate the DFG-in conformation again.

  1. Paragraph 2.1., in Figure 2A R4 and R6 are labelled whereas in the text R3 and R5 are mentioned. Are there any mistakes?

No – this was no mistake, but not explained well. In Figure 2B, the residues arginine R4 and arginine R6 are labelled. In the text, the positions relative to the phosphoacceptor are listed (positions -3 and -5). This is explained in the text now.

  1. Paragraph 2.2, what is aF loop?, what is its role for the kinase structure and activity?

The ‘aF aG loop’ is just the loop connecting the aF and aG helices. I am not aware of any particular role of this loop in other kinases. It is indicated in yellow in the AKT2 model in Figure 2B now.

  1. Paragraph 2.2. Is it possible to discuss a bite more about LIMK dual specificity (Ser/Tyr)? Is it common for other kinases? How may this feature be explained?

Several sentences have been added to discuss the dual specificity. MEK6 was mentioned as an example. The LIMK dual specificity is explained with the ‘rock-and-poke’ mechanism. 

  1. Paragraph 2.3., please mention the structures of LIMK LIM and PDZ structures obtained by NMR and deposited in the PDB. Maybe comments about these structures are necessary?

The structure models are mentioned now. Both of them have not been published yet.

  1. In paragraph 2.3., several claims are not exact. Edwards and Gill (1999, JBC 274, 16, 11352) showed that LIMK LIM domains negatively regulate its kinase activity, and Hiraoka et al (1996, FEBS Letters, 399, 117) showed an interaction between N and C-terminal domains of LIMKs. Several partners of LIMKs involving their LIM and PDZ domains have been identified: Par3 interacts with the LIM domain of LIMK2 (Chen and Macara, 2006, JCB 172,671), Nurr1 interacts with the LIM domain of LIMK1 (Sacchetti et al., NAR 2006, 34, 19, 5515-5527), the SecPH domain of Nf1 interacts with the PDZ of LIMK2 (Vallee et al, 2012 PLOS One, 7, 10 e47283), Nischarin interacts with the PDZ domain of LIMK1 (Ding et al., MCB 2008, 28, 11, 3742), …

Paragraph 2.3 has been completely remodelled – all suggested interactors are described and cited now.

Reviewer 2 Report

Chatterjee at al. have written a clear and comprehensive review about the structural features of LIMK proteins catalytic activities and regulations and their impact on the process of drug discovery. The field was in need of such a review because many structures of LIMK1 or 2 kinase domains have been resolved in complex with different inhibitors or cofilin substrate thereby allowing a detailed understanding of their catalytic mechanism.

I have a few comments which will improve the manuscript.

-The authors should do in paragraph 1 what they did in paragraph 2: explain in a first specific sub paragraph the most important features of kinase structures and present the important and conserved elements of these structures: DFG motif which can be in and out and activation loop, gatekeeper residue, VAIK lysine motif, aC helix, HRD motif histidine, G-rich loop…

Once defined, these elements should then be shown in inactive and active LIMK conformations.

-For a better understanding, aC glutamate (E384), VAIK lysine K368 and D478 from DFG should be labelled in Figure 1.

-In Figure 2, for a better understanding, the aF-aG loop should be labelled in a specific color.

-In 2.3 paragraph, the authors discuss about the possible inhibitory role of LIM and PDZ domains on the kinase activity of LIMKs as described for other kinases. They exclude this possibility and interactions between LIM/PDZ domains and kinase domain. They should not be so assertive because Hiraoka et al. (1996) showed an interaction between these domains and Arber et al. (1998) bring it up to explain the higher kinase activity of LIMK kinase domain when LIM and PDZ domains are deleted.

-Li et al. (2006) have shown that LIMKs interact with HSP90 via their kinase domain and that this interaction promotes LIMKs dimerization and transphosphorylation thereby stabilizing them. This result should be cited and discussed in light of LIMK structure and inhibitors development.

Author Response

Dear reviewer --

Thank you very much for your comments / suggestions! This has allowed us to correct several mistakes and to add some important aspects of LIMK biology. Please find below a detailed reply to your review.

Kind regards

Chatterjee et al.

Reviewer 2

This manuscript by Chatterjee et al. provides a review of structural and pharmacological studies of LIM kinases, a group of Ser-Thr kinases that have important roles in regulation of the actin cytoskeleton in animal cells. I am not aware of a recent review on the topic, and given recent progress in the area this is a timely manuscript that will be of broad interest. Overall the review does a nice job covering the key work in this area, though as detailed below, prior work in the area of LIMK regulation merits discussion. I do have a few suggestions for improving the manuscript prior to publication.

Major points:

  1. Section 2.2: The “rock and poke” mechanism for LIMK phosphorylation of cofilin could be elaborated upon with a figure panel. Though Fig 2B shows the LIMK-CFL complex, the “rocking” mechanism is only indicated by a curved arrow. Could an overlay of the two conformations be shown, or perhaps a side-by-side comparison? Does the positioning of the phosphoacceptor sequence differ between the different LIMK-CFL structures?

We have modified Figure 2D accordingly. A second CFL1 molecule was added to the figure demonstrating the rocking movement. Also, differences between the phosphoacceptor positions are obvious now.

  1. Section 2.3 should discuss prior studies showing autoinhibition of the LIMK catalytic domain by the non-catalytic N-terminus. Despite what is stated in the manuscript, the N-terminal region reportedly associates directly with the catalytic domain and inhibits its activity (Hiroaka et al., FEBS Lett 1996, 399, 117-121; Nagata et al., Biochem J, 1999, 343, 99–105; Edwards & Gill, 1999, 274, 11352–11361). It is also worth noting that TGFBRII, which inhibits LIMK, appears to interact with the first LIM domain. Finally, multiple kinases promote LIMK activity by phosphorylation of the linker connecting the N-terminal region and the kinase domain (Cell Cycle, 2012, 11, 296-309; EMBO J, 2006, 25, 713-26).

The entire section 2.3 was re-organized. The suggested LIMK interactors and regulators are described and cited accordingly.

Minor points:

  1. Almost all other Ser-Thr kinases have a catalytic loop Lys residue (in the HxDxKxxN motif), but this reside is Asn in LIMKs. Can the authors comment on the likely significance of this unique feature of LIMKs?

This unusual residue in the HRD+2 position is now mentioned in section 1.2. The impact is not clear – we hypothesize there will be an impact on the apparent Km for ATP.

  1. In Section 1.3, it is incorrectly stated that BMPRII was reported to phosphorylate LIMKs (ref 15). In fact, that paper reported that BMPRII inhibited LIMKs and blocked their interaction with PAK4.

This mistake was corrected.

  1. Section 2.1, 1st paragraph: “Substrate loop” is unusual nomenclature and suggests a specific structural feature present in substrates. “Grove” should be “groove”.

‘Substrate loop’ was changed to ‘phosphoacceptor loop’. The phosphoacceptor loop was defined as the phosphoacceptor residue plus flanking residues to avoid the impression this was a specific structural feature. ‘Grove’ was made ‘groove’.

  1. Section 2.1, 2nd paragraph: “ALK2” should be “AKT2”

This mistake was corrected.

  1. Discussion of conventional kinase substrate interactions in section 2.1 doesn’t include main chain interactions between the activation loop and the substrate peptide – one of the unique features of the LIMK-CFL complex is that these interactions do not occur.

The main chain interactions were added to the description of the AKT2-substrate interactions.

  1. In section 2.2 it is stated “…specificity is here not defined by the length of the side chain (as discussed above for conventional kinases), but by steric shielding of the hydroxy group.” What is meant by “steric shielding” in this context? This implies that the hydroxyl is somehow being blocked from accessing the catalytic center. It’s not clear to me how this relates to the LIMK mechanism. Perhaps the authors could elaborate on this point.

It is difficult to explain why the serine phosphoacceptor is preferred over threonine. For conventional kinases, the DFG+1 residue plays a role – this is discussed in the manuscript now. Another possibility is that the threonine methyl prevents the threonine hydroxy from attacking the gamma phosphate when poking into the active site. We tried to explain this better in the text now.

  1. Given that there are no reported substrate competitive LIMK inhibitors and the authors suggest they are not even worth pursuing, perhaps section 3.3 could be removed?

The idea was that while it is scientifically interesting and elegant to target the docking interface, it is more difficult than developing an active site binder. The paragraph was re-phrased.

  1. Section 3.5: A few observations regarding differences between LIMK1 and LIMK2 are worth noting. Early studies that mutations flanking Ser3 in the cofilin N-terminus differentially impacted phosphorylation by LIMK1 and LIMK2 (Amano et al. Biochem J, 2001, 354, 149–159), indicating that they have different substrate specificity. Though the physiological relevance is not clear, this does suggest differences in the catalytic center that suggest the feasibility of targeting one isoform selectively. Also worth noting is that LIMK1 and LIMK2 have different Gly-loop sequences (GKGFFG and GKGCFG, respectively). The presence of a unique Cys residue in LIMK2 is potential relevant to the development of isoform-specific inhibitors.

Both suggestion were included into Section 3.5 – thank you.

Reviewer 3 Report

This manuscript by Chatterjee et al. provides a review of structural and pharmacological studies of LIM kinases, a group of Ser-Thr kinases that have important roles in regulation of the actin cytoskeleton in animal cells. I am not aware of a recent review on the topic, and given recent progress in the area this is a timely manuscript that will be of broad interest. Overall the review does a nice job covering the key work in this area, though as detailed below, prior work in the area of LIMK regulation merits discussion. I do have a few suggestions for improving the manuscript prior to publication.

Major points:

  1. Section 2.2: The “rock and poke” mechanism for LIMK phosphorylation of cofilin could be elaborated upon with a figure panel. Though Fig 2B shows the LIMK-CFL complex, the “rocking” mechanism is only indicated by a curved arrow. Could an overlay of the two conformations be shown, or perhaps a side-by-side comparison? Does the positioning of the phosphoacceptor sequence differ between the different LIMK-CFL structures?
  2. Section 2.3 should discuss prior studies showing autoinhibition of the LIMK catalytic domain by the non-catalytic N-terminus. Despite what is stated in the manuscript, the N-terminal region reportedly associates directly with the catalytic domain and inhibits its activity (Hiroaka et al., FEBS Lett 1996, 399, 117-121; Nagata et al., Biochem J, 1999, 343, 99–105; Edwards & Gill, 1999, 274, 11352–11361). It is also worth noting that TGFBRII, which inhibits LIMK, appears to interact with the first LIM domain. Finally, multiple kinases promote LIMK activity by phosphorylation of the linker connecting the N-terminal region and the kinase domain (Cell Cycle, 2012, 11, 296-309; EMBO J, 2006, 25, 713-26).

Minor points:

  1. Almost all other Ser-Thr kinases have a catalytic loop Lys residue (in the HxDxKxxN motif), but this reside is Asn in LIMKs. Can the authors comment on the likely significance of this unique feature of LIMKs?
  2. In Section 1.3, it is incorrectly stated that BMPRII was reported to phosphorylate LIMKs (ref 15). In fact, that paper reported that BMPRII inhibited LIMKs and blocked their interaction with PAK4.
  3. Section 2.1, 1st paragraph: “Substrate loop” is unusual nomenclature and suggests a specific structural feature present in substrates. “Grove” should be “groove”.
  4. Section 2.1, 2nd paragraph: “ALK2” should be “AKT2”
  5. Discussion of conventional kinase substrate interactions in section 2.1 doesn’t include main chain interactions between the activation loop and the substrate peptide – one of the unique features of the LIMK-CFL complex is that these interactions do not occur.
  6. In section 2.2 it is stated “…specificity is here not defined by the length of the side chain (as discussed above for conventional kinases), but by steric shielding of the hydroxy group.” What is meant by “steric shielding” in this context? This implies that the hydroxyl is somehow being blocked from accessing the catalytic center. It’s not clear to me how this relates to the LIMK mechanism. Perhaps the authors could elaborate on this point.
  7. Given that there are no reported substrate competitive LIMK inhibitors and the authors suggest they are not even worth pursuing, perhaps section 3.3 could be removed?
  8. Section 3.5: A few observations regarding differences between LIMK1 and LIMK2 are worth noting. Early studies that mutations flanking Ser3 in the cofilin N-terminus differentially impacted phosphorylation by LIMK1 and LIMK2 (Amano et al. Biochem J, 2001, 354, 149–159), indicating that they have different substrate specificity. Though the physiological relevance is not clear, this does suggest differences in the catalytic center that suggest the feasibility of targeting one isoform selectively. Also worth noting is that LIMK1 and LIMK2 have different Gly-loop sequences (GKGFFG and GKGCFG, respectively). The presence of a unique Cys residue in LIMK2 is potential relevant to the development of isoform-specific inhibitors.

Author Response

Dear reviewer --

Thank you very much for your comments / suggestions! This has allowed us to correct several mistakes and to add some important aspects of LIMK biology. Please find below a detailed reply to your review.

Kind regards

Chatterjee et al.

Reviewer 3

Chatterjee at al. have written a clear and comprehensive review about the structural features of LIMK proteins catalytic activities and regulations and their impact on the process of drug discovery. The field was in need of such a review because many structures of LIMK1 or 2 kinase domains have been resolved in complex with different inhibitors or cofilin substrate thereby allowing a detailed understanding of their catalytic mechanism.

I have a few comments which will improve the manuscript.

-The authors should do in paragraph 1 what they did in paragraph 2: explain in a first specific sub paragraph the most important features of kinase structures and present the important and conserved elements of these structures: DFG motif which can be in and out and activation loop, gatekeeper residue, VAIK lysine motif, aC helix, HRD motif histidine, G-rich loop…

The respective sub paragraph was added.

Once defined, these elements should then be shown in inactive and active LIMK conformations.

Details of the active and inactive LIMK conformations are depicted in Figure 1D to 1G now – a new panel was added to compare the DFG conformations.

-For a better understanding, aC glutamate (E384), VAIK lysine K368 and D478 from DFG should be labelled in Figure 1.

The respective residues are labelled in Figure 1C now.

-In Figure 2, for a better understanding, the aF-aG loop should be labelled in a specific color.

The aF aG loop is labelled in yellow now.

-In 2.3 paragraph, the authors discuss about the possible inhibitory role of LIM and PDZ domains on the kinase activity of LIMKs as described for other kinases. They exclude this possibility and interactions between LIM/PDZ domains and kinase domain. They should not be so assertive because Hiraoka et al. (1996) showed an interaction between these domains and Arber et al. (1998) bring it up to explain the higher kinase activity of LIMK kinase domain when LIM and PDZ domains are deleted.

Paragraph 2.3 is completely re-shuffled now. The influence of the N terminus on LIMK kinase activity is discussed in more detail now. The respective literature is cited.

-Li et al. (2006) have shown that LIMKs interact with HSP90 via their kinase domain and that this interaction promotes LIMKs dimerization and transphosphorylation thereby stabilizing them. This result should be cited and discussed in light of LIMK structure and inhibitors development.

The LIMK interaction with HSP90 is described in paragraph 2.3 now.

Round 2

Reviewer 1 Report

The authors have answered all the questions asked, and modified the text accordingly.

This paper is perfect for publication.